# Neurally Guided Genetic Programming for Turing Complete Programming by Example

## Abstract

The ability to synthesise source code from input/output examples allows non-experts to generate programs, and experts to abstract away a wide range of simple programming tasks. Current research in this area has explored neural synthesis, SMT solvers, and genetic programming; each of these approaches is limited, however, often using highly specialised target languages for synthesis. In this paper we present a novel hybrid approach using neural networks to guide genetic programming (GP), which allows us to successfully synthesise code from just ten I/O examples in a generalised Turing complete target language, up to and including a sorting algorithm. We show that GP by itself is able to synthesise a set of simple programs, and show which hints (suggested lines of code for inclusion) are of most utility to GP in solving harder problems. Using a form of unstructured curriculum learning, we then demonstrate that neural networks can be used to determine when to make use of these high-utility hints for specific I/O problems and thus enable complex functions to be successfully synthesised. We apply our approach to two different problem sets: common array-to-array programs (including sorting), and a canvas drawing problem set inspired by So & Oh (2018).

## 1 Introduction

The ability to synthesise source code from examples, in which a source code implementation of a function is created based on one or more demonstrations of input-output mapping, is a fundamental question in machine learning. We specifically study this question in the form of scenarios where large corpora of existing code are not available (e.g., human-written programs in open-source repositories). The immediate applications of this would allow non-programmers to generate programs, or experts to abstract away trivial coding tasks. In addition, from a machine learning perspective, it allows complex functions to be generated in a symbolic and human-readable form – which can be subjected to a wide range of static analysis tools to model generality or correctness.

To date this challenge has been studied using neural-network-driven synthesis, genetic programming, and SMT solvers. However, at present these approaches are significantly constrained in the complexity of the target language in which code is synthesised. Neural synthesis for example, such as the DeepCoder architecture (Balog et al., 2017; Zohar & Wolf, 2018), shows success on simple domain-specific languages, but the search space of more realistic Turing-complete languages is vast by comparison and is unlikely to be representable in a neural network on current or near-future hardware. Genetic programming, meanwhile, is limited by our ability to specify a fitness function which can successfully navigate to a solution for a particular problem – in the highly irregular and often flat fitness landscape of program space (Kinnear, 1994; Renzullo et al., 2018). SMT solvers by comparison lack the analytical power to handle loops without human guidance, constraining their applicability (Srivastava et al., 2010a; So & Oh, 2018; Srivastava et al., 2010b).

In this paper we examine code synthesis from examples for a Turing-complete language which can be cross-compiled into C/Java. We use just 10 input/output examples to describe each problem for which we need to synthesise a matching function (i.e., providing unsorted and sorted arrays of integers to describe sorting); because our target language yields a total search space size of $5 * 10^{119}$ possible program permutations, scalability of the synthesis technique is crucial. In this context, we use a novel combination of genetic programming (GP) and neural networks (NNs); GP is used to navigate within program space from a given starting point using a general-purpose fitness function,

while NN methods are used to provide prediction of high-level features which help guide the GP to higher-probability success areas in which to search. In essence this technique allows the NN to model only highly abstract features of program space, allowing it to scale to vast program search spaces, while the GP can then incrementally traverse program space from an NN-derived starting point to a correct solution. We bootstrap this process by using an unstructured form of curriculum learning, in which successfully-found functions are used as seed programs to generate synthetic corpora on which to train new neural networks, leading to further high-utility source code feature predictions for new problems. Initially this curriculum learning is based on programs that can be successfully found using GP alone with our generic fitness function, which then allows us to synthesise more complex programs using NN inference.

Our key results demonstrate that GP is able to solve simple synthesis problems unaided, and that synthetic corpora generated from these problems allow popular neural network architectures to identify high-utility search hints on more complex problems to guide the genetic programming search. In one of our problem sets, this has the effect of allowing the framework to successfully synthesise 7 of the 10 programs that had never been found by GP alone, and among other programs moves success rates from 38% to 55%. All of our source code is provided online (pending de-anonymisation).

## 2 RELATED WORK

Code synthesis from I/O examples has been studied using three major approaches: deductive solvers; neural networks (NNs) with search; and genetic programming (GP). For code synthesis in a Turing-complete language, deductive solvers have yet been shown to operate well with loop-based flow control operators (although frameworks which manually define any non-linear program flow can yield good performance (So & Oh, 2018)). Neural synthesis, by comparison, is limited by how much of program space for a general-purpose target language can be captured in a NN model, while GP is limited by the difficulty of deriving a fitness function to navigate to a solution (Renzullo et al., 2018). In the remainder of this section we focus on NN and GP approaches in more detail.

**Neural synthesis** Neural synthesis works by training a neural network on a sub-sample of the entirety of program space for the target language (e.g., sampling at a uniform interval or at random). When presented with a new problem as an I/O example, the neural network will then be asked to predict which lines of code (or particular operators) are likely to be present in the solution based on similar I/O transforms observed in the training set from the above sub-sample. The system will then perform an exhaustive search of program space to fill in the remaining (non-predicted) features. Notable examples here include DeepCoder and RobustFill, among others (e.g., Balog et al. (2017); Zohar & Wolf (2018); Devlin et al. (2017); Chen et al. (2019); Singh & Kohli (2017))

The key limitation to this approach is that it must be able to train on a detailed enough sub-sample of program space, and store this sample inside a neural network, to make meaningful predictions on program features for unseen problems. While this works for highly simplified languages (Deep-Coder, for example, has no loop operators (Balog et al., 2017)), the search space size of a Turing-complete language is astronomical by comparison. If we consider that the capacity of a feed-forward ReLu neural network to differentiate between classes (in our case programs), termed its VapnikCher-vonenkis (VC) dimension, at best grows as a linear function of $w * L * log(w)$ where $w$ is the total number of weights and $L$ the number of layers (Bartlett et al., 2019), it is unlikely that a neural network on current hardware would be able to represent a useful sub-sample of possible program permutations yielded by the search space of a general-purpose language.

**Genetic programming** GP relies on iterative travel through program space from a starting point (often an empty program) to the solution, guided by a fitness function (Vanneschi & Poli, 2012; Taleby Ahvanooey et al., 2019). The field has a long history (Forsyth, 1981) but still shows results (Miranda et al., 2019) that are competitive with neural networks (Ain et al., 2020), and an ability to tackle complex problems mixing diverse datatypes (Pantridge & Spector, 2020).

Unlike neural synthesis, a GP approach does not need to encode the entirety of program space in a model, and so can in theory work in a scalable fashion on high dimensional search spaces as long as a fitness function is provided which can guide the search incrementally towards a solution. The key problem with GP for code synthesis is that large areas of program space are difficult to navigate,

exhibiting large plateaus of neutrality (no behavioural change despite significant code change) and highly irregular responses to code change (jagged fitness landscape) (Renzullo et al., 2018; Kinnear, 1994). A fitness function may also fail to capture higher level properties, for example "all outputs are even" or "all values from the input are repeated in the output", which may otherwise be identified by a neural network.

**Combining neural network prediction with genetic programming**    Given the limitations of both NNs and GP in themselves, we hypothesise that the combination of the two techniques may provide the best features of both while mitigating their respective limitations in the context of code synthesis in a Turing-complete language with a very large program search space.

While a NN cannot feasibly model all of program space, and so cannot be expected to predict each line to be synthesised, it does have the potential to predicting a small number of higher-level features which only require a very limited internal model of program space. This can be combined with a GP search process which fixes these lines in place to constrain the search area, and can use this partially-constructed program in combination with one or more generic fitness functions to guide the search to a successful result. We further find that we can use successfully-found programs to generate synthetic training sets for a NN focused on that area of program space, which then leads to further high-value predictions of likely features for more complex programs.

## 3    METHODOLOGY

Our system accepts up to ten I/O examples from a user and synthesises the source code of a function which converts those inputs to their corresponding outputs. We use 10 examples as a number which a user may be willing to input while representing lower effort than writing the function manually. If our system fails to synthesise source code for a given problem, the user can specify a simpler but related I/O problem; the successful synthesis of this simpler problem can then lead to subsequent success on the more complex problem. As a target programming language for synthesis, we use a Turing-complete language (previously used by the authors of (Wild & Porter, 2019)) which can be cross-compiled directly into C/Java/Python. The language features primitive loop operators, variable declarations, and conditional branch operators; a full listing of its operators is given in appendix A.1.

We use a combination of GP guided by NN prediction to approach this problem, and show successful synthesis of a wide range of programs including sorting. In this paper we study three specific parts of the above overall framework: (i) the success rate of GP by itself of finding programs in our problem sets; (ii) the effect of different *hints* (suggested lines of code) on the success rate of GP; (iii) the ability of a NN to predict high-utility hints, after being trained on a synthetic corpus derived from high-success program finds from the GP alone.

In the remainder of this section we describe our problem sets, genetic programming and NN implementations in more detail, then present the set of experiments that we conduct.

### 3.1    SPECIFICATION OF PROBLEM

We use two different problem sets which represent human-useful programs of the kind that may be input into our system. Our first problem set is array-to-array programs such as extract even numbers, append arrays, or sort. Our second problem set is provided with an image to draw on a canvas and must synthesise the program which draws that image (this problem set is taken from So & Oh (2018)).

Each problem within each problem set is presented to the system as a set of 10 IO examples. These are generated for each problem by feeding in 10 inputs and corresponding outputs of the form the problem requires (either a randomly generated input array and integer, or a canvas size). These 10 inputs are randomly generated from a fixed seed, to reduce internal variability between tests, allowing more accurate evaluation of the changes made by alterations to parameters or by guidance to the GP. This is designed to represent an unbiased input set.

### 3.2 GENETIC PROGRAMMING

Our GP algorithm creates a population of 2,000 programs, each one the result of crossover and mutation from among the 10 highest-ranked parents of the previous generation (or, for the first generation, mutations of the empty program). Each program in a population is then ranked using a general-purpose fitness function which was experimentally found to be good at locating programs from various search space starting points. We integrate novelty search (Lehman & Stanley, 2010; Doncieux et al., 2019) as part of our fitness function to avoid falling into the same local minima repeatedly, which was found to further boost search success. Once all programs have been ranked, a new generation of 2,000 members is then created, repeatedly, until the target program is found or the system reaches a maximum number of generations (3,000) and reports failure.

Our fitness function, in detail, is formulated as $error * novelty\_penalty$. The value for $error$ is calculated as the number of elements in the output of a candidate function which do not match those expected for the corresponding input; this value is normalised into the range $[0.0, -1.0]$ by dividing it by the corresponding error which would have been produced by an *empty* output for that input.

The value for $novelty\_penalty$ is calculated by first extracting the highest-fitness member of a population, and removing all lines of code which do not contribute to its behaviour. This program is stored as a *repulsor* which indicates how well-explored a given region of program space is. When calculating the fitness of a new program, each repulsor adds a multiplicative value depending on the distance $D$ of that program from the repulsor. $D$ itself is calculated by examining each line in the first program and calculating how far away (in lines of code) the same line is in a second program. Each repulsor adds $max(0, 1 - D/15)$ to $novelty\_penalty$, such that multiple repulsors can exist in the same area of program space, leading to stronger avoidance of those areas.

For each new generation during the GP search, parents are chosen using tournament selection with tournament size of 10 (Miller & Goldberg, 1995). Crossover occurs by taking the first half of the first parent, and appending the second half of the second parent (syntactically flawed offspring are accepted into the population but receive minimal fitness when evaluated). Following crossover, we apply a single mutation to a program with a probability of 0.35. Following each mutation a random boolean value is selected, and if true another mutation is applied, to a maximum of 8 mutations.

Each mutation takes one of the following forms, selected uniformly at random:

1. Insert: this inserts a random line from all possible lines available in the language, and will delete last lines of program if program is already at maximum length. It automatically adds an ENDBLOCK operator if a flow-control operator was inserted.

2. Delete: sets a random line to NO-OP.

3. Mutate: changes a random operator/parameter on a line, ensuring the line remains valid.

4. Swap: exchanges position of any two lines of the program.

In all of the above, the parameter values relating to $D$ and mutation probabilities were chosen as values found to experimentally work well.

### 3.3 NEURAL NETWORK DESIGN

While no programs have yet been successfully synthesised by our system, we initially rely on GP as above to locate simpler programs. Once at least one program has been located it is added to our set of successfully found programs $S_F$. We then augment GP with a neural network which provides hints of one or more likely lines of code for a given unsolved problem, which correspond to likely areas of program space in which the GP will search (rather than the GP starting its search from an empty program). Our NN is trained on a synthesised training set of programs; the programs in this training set are generated at random but must exhibit some of the program features present in programs in $S_F$. The intuition here is that the neural network will thus learn to recognise which program features are likely to be present for unseen I/O examples, based on program features known to be useful in other programs requested by users, which are then useful to guide the GP search.

Each synthesised training corpus based on $S_F$ has 20,000 programs for training and 2,000 for testing. The 20,000 programs are divided into 10,000 which do have a particular program fragment

of interest, and 10,000 which do not have that fragment, with the NN trained to determine whether or not a given I/O example is likely to have that fragment in the corresponding implementation program. Each program in each set of 10,000 is assured to be distinct in functionality from every other program in the set, tested on its behaviour with respect to a fixed 10 I/O examples. Each new program in a training set is generated by selecting uniformly at random two programs already accepted, applying a crossover, then applying between 1 and 8 mutations (as described above in the GP section) while assuring that the fragment of interest still exists.

We use a range of NN architectures to study which ones work best in each problem set. Specifically we use a feed-forward NN in both problem sets, as a shared baseline; in our array-to-array problem set we then also use an LSTM-based recurrent network, and in our canvas problem set we use a CNN. The details of all network architectures are described in appendix A.3.

## 3.4 EXPERIMENT SETUP

We use three different experiments to study each element of our approach. The details of each experiment are given below, while the next section presents the results.

**Experiment 1: Exhaustive Fragment Evaluation**    In our first experiment we examine the baseline performance of GP alone on our two problem sets, then study the effect of each possible source code hint that can be given for each problem (in terms of its effect on success rate).

For the latter we examine the set of 1 or 2-line code fragments which can be cleanly isolated (with no dependencies) in the ground truth solution to each problem. We then run a GP pass with the code fragment as a forced requirement, such that any program produced by the GP which does not include them automatically receives a penalty fitness of -10,000. Each experiment in this series is repeated 30 times to account for the inherent stochasticity in the GP process.

These results demonstrate the kinds of problem that can be solved using GP alone, and the extent to which a GP algorithm can have its probability of finding a solution increased by constraining its sampled programs to contain certain lines of code – which help identify the highest utility hints that a NN can seek to find.

**Experiment 2: Fragment Recognisability by Neural Network**    Having established that code fragments can be used to improve GP find rates (including from zero to non-zero find rates), our second experiment studies the use of synthetic NN training corpora based around these fragments – and the extent to which NNs can successfully predict these features in unseen problems to assist GP.

For this experiment we need to assume that some programs have already been found by the GP alone, and useful fragments identified, from which to automatically synthesise NN training sets as described above.

For our array-to-array problem set, we select all programs which have find rates of 90% and above using the GP alone from experiment 1. For our canvas problem set, we select the single highest find-rate program using the GP alone for each 'class' of problem identified by So & Oh (2018) (such as 'triangles'). These programs represent those which are most likely to have been found first without the aid of the NN. From within the source code of these programs we select a set of individual fragments from which to generate our synthetic NN training corpora. These fragments are chosen using high-utility fragments identified from experiment 1, and augmented with further fragments of interest that were manually chosen to gain a wider coverage of fragment predictability by a NN.

Altogether, these experiments indicate how well NNs can predict the presence of different program features based on our synthetic training sets – a mechanism by which the solution to easy (high-find-rate) problems can be used to find solutions to hard (low-find-rate) problems.

**Experiment 3: Success of chosen fragments**    In the third experiment we test the success of our GP algorithm when guided by NN-predicted source code fragments using the trained NNs from experiment 2, using all problems in each problem set. To do this we apply a simple selection process, choosing uniformly at random from among all fragments which were estimated by an NN to have a probability of $>= 0.5$ presence in a program, and provide one such fragment to the GP process. This shows how a full neurally-guided GP process would perform in our end-to-end system.

| Problem | Baseline | Maximum | Best Operators |
|---------|----------|---------|----------------|
| Append | 0% | 27% | Var=Literal 1; Addition |
| Cumulative Abs Sum | 0% | 3% | Loop; Read |
| Keep Evens | 0% | 7% | Var=Literal 2; Make Array |
| Retain First Half | 0% | 13% | Var=Literal 2; Divide |
| Reverse | 58% | 80% | Var=Literal 1; Make Array |
| Shift Right | 0% | 13% | Var=Literal 1; Loop |
| Shift Right Lossy | 84% | 80% | Var=Literal 1 |
| Sort (Bubblesort) | 0% | 0% | (None) |
| Parallelogram | 7% | 30% | Var=Literal 2; Divide |
| Mirrored Hollow Parallelogram | 13% | 60% | Var=Literal 2; Divide |
| Hollow Right Triangle | 87% | 90% | Var=Literal 1; Subtract |
| Hollow Mirrored Right Triangle | 63% | 93% | Var=Literal 1; Subtract |
| Inverted Isosceles Triangle | 46% | 23% | Var=Literal 2 |
| Trapezoid | 7% | 10% | Var=Literal 1 |

Table 1: Find rates for guided Genetic Programming Algorithm with forced inclusion of code fragments from ground truth. Baseline is unguided GP. Maximum is single best performing fragments. Best operators are those used in the highest-scoring fragment (first if tied) (n=30 per fragment, percentage success)

## 4 EVALUATION

Our evaluation was conducted on Tensorflow 1.14, Python 3.6.9, Java OpenJDK 11.0.6. Our source code will be made available in camera-ready version

### 4.1 EXPERIMENT 1: GP BASELINE AND ITERATIVE REQUIREMENT SEARCH

In this experiment we first determine the baseline find rate of the GP with no assistance, and also select a number of problems to study by supplying certain subsets of the lines as guidance to the GP. This is done by fully running 30 GP search repeats with each valid (as described above) 2-line fragment from the ground truth. From the first corpus we selected 8 problems, from the second we select 6. From the first corpus we select mostly low-find-rate problems, to study which form of fragments would be useful to provide to achieve success in the GP, while in the second corpus we select a more representative sample, to study how constraint-forcing behaves in general. We select two problems of the same class from the 2nd corpus (right triangles), to ensure that similar fragments provide similar results in similar circumstances (to give confidence in generality of these results). The baseline success, best find rate increase, and best fragment's operators for selected problems are presented in Table 4.1.

Here we clearly see that forcing the inclusion of even the simplest code elements (one or two lines of the ground truth) into the GP's population allows the GP to find previously unsolvable problems. We can also see that fragments containing arithmetic operators (especially literal assignment) appear to have a stronger influence on success, possibly as they are harder to find, as their effects are far more subtle and complex than, say presence of a loop operator. These should therefore be studied as high-utility candidates for deployment into GP processes we wish to guide in the future.

We also note that some examples show a *decrease* in success rate (e.g. "Shift Right Lossy"). While no fragment reduced find rates to zero, we speculate that in some cases the provision of a fragment places the GP into an area of program space from which it is harder to reach the solution using our general-purpose fitness function (for example, meaning that this point in program space has larger regions of neutral landscape around it which are harder to traverse over).

A full breakdown of all baseline GP find rates is presented in Appendix A.4, with fragments and their successes presented in Appendix A.5.

| Fragment | FFNN | FFNN Test | RNN | RNN Test |
|---|---|---|---|---|
| Add | 58 % | 62 % | 58 % | 62 % |
| +1 Offset Loop | 74 % | 67 % | 71 % | 66 % |
| Length -1 Loop | 76 % | 66 % | 77 % | 65 % |
| Literal (2) | 76 % | 63% | 71 % | 60 % |
| Loop | 97 % | 78% | 98 % | 79 % |
| Loop+Conditional | 72 % | 62% | 70 % | 59 % |
| Loop+Read | 65 % | 61% | 62 % | 58 % |
| Nonstandard Array | 81 % | 68% | 80 % | 66 % |
| Read | 76 % | 63% | 60 % | 58 % |
| Subtract | 61 % | 56% | 62 % | 56 % |
| Average | **74** % | **65** % | 71 % | 63 % |
| **Fragment** | **FFNN** | **FFNN Test** | **CNN HU** | **CNN Test** |
| Add | 58 % | 67 % | 78 % | 86 % |
| Conditional | 61 % | 57 % | 64 % | 85 % |
| Half | 61 % | 62 % | 87 % | 87 % |
| Half Loop | 65 % | 60 % | 82 % | 87 % |
| Half Loop Depend | 70 % | 59 % | 82 % | 90 % |
| Loop Conditional | 64 % | 58 % | 57 % | 85 % |
| Loop Draw | 65 % | 58 % | 70 % | 85 % |
| Loop Loop | 73 % | 57 % | 83 % | 75 % |
| Loop Loop Subtract | 60 % | 55 % | 64 % | 77 % |
| Draw Draw | 52 % | 52 % | 74 % | 71 % |
| Average | 63 % | 58 % | **74** % | **83** % |

Table 2: Percentage accuracy of two neural networks on two corpora, with regards to ability to recognise presence of a code fragment within the source code of the function whose IO mapping they are receiving as feature inputs. First NN architecture is a common FFNN implementation. Accuracy is only that of the NNs which show success on known-ground-truth seed fragments (as described in methodology, others discarded from results). Testing accuracy is accuracy on the 2,000 synthetic testing programs. Full description of fragments in Appendix Tables 24 and 25 (n=30)

## 4.2 Experiment 2: Fragment Recognisability

In this experiment we start with selected high-success seed programs from our first experiment. For array-to-array problems, these are all programs with a 90% find rate or better via GP alone, while for our canvas drawing problem set, these are the single highest-find-rate program in each category such as 'triangles'. We then take 10 source code fragments present in these seed programs which exhibit a positive effect of find rates, and use these fragments to generate completely synthetic training corpora for neural networks trained to predict whether or not an I/O example will include a particular fragment in its source code solution.

Table 4.2 shows how effective our different trained neural networks architectures are at then correctly predicting the presence of these fragments in the I/O examples from our two problem sets (which are not part of the synthetic training sets).

This data show significant variance of prediction success (from 58% up to 97%) but overall shows that our neural networks do exhibit the ability to accurately predict that a particular source code fragment will exist in the solution to a given I/O problem – even though these neural networks are trained on entirely synthetic data.

Examining the different neural network architectures, our baseline feed-forward NN shows success in both problem sets and so is able to act as a viable generalist model. Our recurrent network, used in the array-to-array corpus, shows prediction success that is generally lower than the feed-forward NN. Our CNN meanwhile, used in our canvas drawing corpus, shows some significant gains in prediction success compared to the feed-forward NN, though also shows some lower results (for example on predicting the presence of a loop with a conditional). This suggests that a mixture-of-experts approach may be desirable here, but also that a general feed-forward model is viable.

In practice we would use these trained networks alongside a threshold of prediction when determining which fragments to recommend for inclusion in a GP search; we present this in the following section.

### 4.3 EXPERIMENT 3: CHOSEN FRAGMENT DEPLOYMENT

In this experiment, the fragments from the above NN experiments were employed to guide the GP process, based on the average estimates by the trained neural networks. This demonstrates the efficacy of our end-to-end system in taking successfully-found solutions to easy problems and employing their characteristics, via trained NN predictors, to find solutions to more difficult problems.

We report the success rates here using two different approaches to selecting fragments for the GP. We do this either using a uniform random choice of fragments which had an average presence probability estimate of $>= 0.5$ (termed *Uniform*); or using a fragment which is predicted for the problem being solved but has the lowest prediction rate across all other problems (which ideally therefore may be the most information bearing fragment). For our array-to-array problem set we use both approaches, while for our canvas problem set we see less clarity in lowest prediction rates across all problems and so only use the uniform random style.

| Corpus And Approach | Find Rate (vs Baseline) | Gained | Lost |
| --- | --- | --- | --- |
| 1st (1D Arrays), Uniform | 46 % (38%) | 5 | 1 |
| 1st (1D Arrays), Rarest Preferred | 55 % (38%) | 7 | 0 |
| 2nd (2D Array), Uniform | 39 % (36%) | 4 | 1 |

Table 3: Success of the GP when guided by the fragments selected by the neural network. 'Gained' are problems which have a $> 0\%$ find rate which have a baseline of $0\%$, 'Lost' are those which previously had $> 0\%$ but now have $0\%$. (n=30)

As can be seen in Table 4.3, guiding the GP using this process produces notable improvements to the overall program synthesis success rates. Average find rates were boosted, and crucially a number of problems became solvable which were not before, with only a single problem in each problem set failing to be solved. This set of newly-solved problems here includes instance in which the bubblesort algorithm, considered by the authors to be the hardest problem in the set, was successfully found.

This improvement was strongest on the approach which selected the estimated-rarest fragment for use, clearly indicating that the fragments are not equivalent in their usefulness. The ideal way in which to extract the most useful fragments for the GP to use, based on those available from NN predictions, therefore remains a topic of future work.

The full set of guided GP results is available in detail in Appendix A.7.

## 5 CONCLUSION

In this paper we have presented a novel combination of genetic programming and neural network prediction to synthesise code from just 10 I/O examples.

Our framework demonstrates the potential to create a system which searches for solutions to problems within a corpus, finds a subset, extracts code fragments from the successes, then trains a neural network to recognise the presence of these fragments and determines which unsolved problems would benefit from NN-guidance – thus boosting GP find rates on a subsequent pass. We demonstrate that this process can render previously unfindable problems findable, and boost overall find rates, including the successful synthesis of bubble sort.

Our approach is scalable to the search space of Turing-complete languages, and has been demonstrated to work successfully in at least two distinct domains using a common target language which can be cross-compiled to C/Java.

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

## A  Appendix

### A.1  Operators of the Language Used

Tables A.1 and A.1 provide lists of all the operators used for the two corpora used in the language employed in the experiments. Language variants are improved in order to allow effective processing of the two corpora's specific problem domains.

| Operator |
| --- |
| Assign Variable To Array |
| Assign Variable From Array |
| Make Array |
| Variable To Literal |
| Add |
| Subtract |
| Multiply |
| Divide |
| Modulo |
| Assign Var from Var |
| Loop |
| Conditional (var > 0) |
| Conditional (var1 == var2) |
| Conditional (var1 > var2) |

Table 4: Operators available for GP, when using the 1st (array-to-array) corpus

| **Operator** |
| --- |
| Assign Variable To Array |
| Assign Variable From Array |
| Make Array |
| Variable To Literal |
| Add |
| Subtract |
| Multiply |
| Divide |
| Modulo |
| Assign Var from Var |
| Loop |
| Conditional (var > 0) |
| Conditional (var1 == var2) |
| Create 2D Array |
| Get 2D Array Size |
| Var to XY Point from 2D Array |
| Set 2D Array to 0 at XY Point |
| Set 2D Array to 1 at XY Point |

Table 5: Operators available for GP, when using the 2nd (2D pattern) corpus

## A.2 PROBLEMS IN CORPORA

Tables 6 and 7 provide a list of all human-provided problems used to test the system across the experiments. These are defined by a source-code implementation in both the custom language used in this paper, as well as in Java. An example of the behaviour of the array-to-array problems (First corpus), has been given, using a fixed sample input, to illustrate the behaviour.

| Problem | Example |
|---|---|
| Abs | $[4, -2, 1, 0, 3, -5] -> [4, 2, 1, 0, 3, 5]$ |
| ArrayLength | $[4, -2, 1, 0, 3, -5] -> [6, 0, 0, 0, 0, 0]$ |
| ArrayToZero | $[4, -2, 1, 0, 3, -5] -> [0, 0, 0, 0, 0, 0]$ |
| CumulativeAbsoluteSum | $[4, -2, 1, 0, 3, -5] -> [4, 6, 7, 7, 10, 15]$ |
| CumulativeSum | $[4, -2, 1, 0, 3, -5] -> [4, 2, 3, 3, 6, 1]$ |
| DivergentSequence | $[4, -2, 1, 0, 3, -5] -> [0, 0, 1, -1, 2, -2]$ |
| FirstElementOnly | $[4, -2, 1, 0, 3, -5] -> [4]$ |
| Identity | $[4, -2, 1, 0, 3, -5] -> [4, -2, 1, 0, 3, -5]$ |
| IndexParity | $[4, -2, 1, 0, 3, -5] -> [1, 0, 1, 0, 1, 0]$ |
| IterativeDifference | $[4, -2, 1, 0, 3, -5] -> [4, -6, 3, -1, 3, -8]$ |
| KeepEvens | $[4, -2, 1, 0, 3, -5] -> [4, -2, 0, 0, 0, 0]$ |
| KeepNegatives | $[4, -2, 1, 0, 3, -5] -> [0, -2, 0, 0, 0, -5]$ |
| KeepOdds | $[4, -2, 1, 0, 3, -5] -> [0, 0, 1, 0, 3, -5]$ |
| KeepPositives | $[4, -2, 1, 0, 3, -5] -> [4, 0, 1, 0, 3, 0]$ |
| Negative | $[4, -2, 1, 0, 3, -5] -> [-4, 2, -1, 0, -3, 5]$ |
| Pop | $[4, -2, 1, 0, 3, -5] -> [4, -2, 1, 0, 3]$ |
| RemoveFirstElement | $[4, -2, 1, 0, 3, -5] -> [-2, 1, 0, 3, -5]$ |
| RetainFirstHalf | $[4, -2, 1, 0, 3, -5] -> [4, -2, 1]$ |
| Reverse | $[4, -2, 1, 0, 3, -5] -> [-5, 3, 0, 1, -2, 4]$ |
| ShiftLeft | $[4, -2, 1, 0, 3, -5] -> [-2, 1, 0, 3, -5]$ |
| ShiftLeftZeroPadded | $[4, -2, 1, 0, 3, -5] -> [-2, 1, 0, 3, -5, 0]$ |
| ShiftRight | $[4, -2, 1, 0, 3, -5] -> [0, 4, -2, 1, 0, 3, -5]$ |
| ShiftRightLossy | $[4, -2, 1, 0, 3, -5] -> [0, 4, -2, 1, 0, 3]$ |
| ShuffleZerosToBack | $[4, -2, 1, 0, 3, -5] -> [4, -2, 1, 3, -5, 0]$ |
| Signum | $[4, -2, 1, 0, 3, -5] -> [1, -1, 1, 0, 1, -1]$ |
| Sort | $[4, -2, 1, 0, 3, -5] -> [-5, -2, 0, 1, 3, 4]$ |
| SquareValues | $[4, -2, 1, 0, 3, -5] -> [16, 4, 1, 0, 9, 25]$ |
| ToIterator | $[4, -2, 1, 0, 3, -5] -> [0, 1, 2, 3, 4, 5]$ |
| Add | $[4, -2, 1, 0, 3, -5], 4 -> [8, 2, 5, 4, 7, -1]$ |
| Append | $[4, -2, 1, 0, 3, -5], 4 -> [4, -2, 1, 0, 3, -5, 4]$ |
| ClipToMax | $[4, -2, 1, 0, 3, -5], 4 -> [4, -2, 1, 0, 3, -5]$ |
| ClipToMin | $[4, -2, 1, 0, 3, -5], 4 -> [4, 4, 4, 4, 4, 4]$ |
| ConstantAddition | $[4, -2, 1, 0, 3, -5], 4 -> [4, 2, 9, 12, 19, 15]$ |
| FillArray | $[4, -2, 1, 0, 3, -5], 4 -> [4, 4, 4, 4, 4, 4]$ |
| GreaterThan | $[4, -2, 1, 0, 3, -5], 4 -> [-1, -1, -1, -1, -1, -1]$ |
| IterateFromStart | $[4, -2, 1, 0, 3, -5], 4 -> [4, 5, 6, 7, 8, 9]$ |
| LessThan | $[4, -2, 1, 0, 3, -5], 4 -> [1, 1, 1, 1, 1, 1]$ |
| MultiplesOf | $[4, -2, 1, 0, 3, -5], 4 -> [0, 4, 8, 12, 16, 20]$ |
| Multiply | $[4, -2, 1, 0, 3, -5], 4 -> [16, -8, 4, 0, 12, -20]$ |
| Subtract | $[4, -2, 1, 0, 3, -5], 4 -> [0, -6, -3, -4, -1, -9]$ |

Table 6: The first corpus of problems, taking either a single array, or an array and an integer. Example provided of the behaviour of each problem, given a standard example input.

| **Problem** |
| --- |
| Square |
| HollowSquare |
| Parallelogram |
| HollowParallelogram |
| MirroredParallelogram |
| MirroredHollowParallelogram |
| RightTriangle |
| HollowRightTriangle |
| MirroredRightTriangle |
| HollowMirroredRightTriangle |
| InvertedRightTriangle |
| HollowInvertedRightTriangle |
| InvertedMirroredRightTriangle |
| InvertedHollowMirroredRightTriangle |
| IsoceleseTriangle |
| HollowIsoceleseTriangle |
| InvertedIsoceleseTriangle |
| HollowInvertedIsoceleseTriangle |
| RectangleWithEmptyTrapezoid |
| InvertedRectangle |
| obtuseTriangle |
| hollowObtuseTriangle |
| mirroredObtuseTriangle |
| mirroredHollowObtuseTriangle |
| invertedObtuseTriangle |
| hollowInvertedObtuseTriangle |
| invertedMirroredObtuseTriangle |
| hollowMirroredInvertedObtuseTriangle |
| VShape |
| Trapezoid |

Table 7: The second corpus, a set of 2D image generation tasks, drawing simple geometric shapes.

### A.3 NEURAL NETWORK ARCHITECTURES

**Feed Forward Neural Network**    The FFNN is a 5 layer structure with 128 nodes per layer, seLu activation. Each layer is connected to all layers below (dense block). Each layer other than the last has a dropout component, with a training dropout rate of $pKeep = 0.75$. The output is a single sigmoidal unit, loss function is mean squared error. Batch size of 32, maximum steps of 512, early stopping after 12 non-progress epochs on validation loss.

**Recurrent Neural Network**    The RNN encodes both the input and output arrays in two architecturally symmetric branches for every example (so 20 branches). Each branch begins by encoding the values in 2 dense layers with 11 nodes, reLu activation, with dropout with $pKeep = 0.75$ after each. After this encoding layer, a layer of 8 LSTM nodes was connected. These LSTM layers represented the end of the two branches. All LSTM outputs are concatenated, along with the input parameters, into a single representation of the IO examples. These are then processed by a set of 2 dense layers with 64 nodes, reLu activation. The output is a single sigmoidal unit, loss function is mean squared error. Batch size of 32, maximum steps of 512, early stopping after 12 non-progress epochs on validation loss.

**Convolutional Neural Network**    Each of the 10 examples of problem output are split into their own branch. Each branch contains a 2D input with width/height equal to the maximum input size, 32. We feed this into a 2D convolutional layer, with stride of 2 and kernel size of 3, reLu activation. We then feed this into a max pooling layer of 2 by 2. We then feed through a second convolutional layer of identical configuration to the first, and a second max pooling layer, again identical. Each

branch then terminates in a single dense layer, 64 nodes, reLu activation. All branches are then concatenated into a single dense layer, 64 nodes, reLu activation. The output is a single sigmoidal unit, loss function is mean squared error. Batch size of 32, maximum steps of 512, early stopping after 12 non-progress epochs on validation loss.

## A.4 PROBLEM FIND RATES AND DESCRIPTIONS

The tables89 are the find rates for the genetic programming algorithm, without any constraints. Note the high degree of variability between problems, with a number from both corpora achieving either a 100% success rate of a 0%.

One remark is that the "Square" problem, which simply requires the entire canvas to be covered (therefore requiring two loops and a write), did not achieve 100% success. This is believed to be due to the genetic algorithm starting with a flawed partial solution, and being unable to move away from this detrimental start due to the nature of the fitness landscape (trapped in local maximum).

## A.5 FULL BREAKDOWN OF FRAGMENTS EVALUATED IN EXPERIMENT 1

Tables 10 to 23 describe each fragment evaluated by the exhaustive fragment testing process. Each fragment is at most two lines, and has no variables which depend on being set in lines outside the fragment (therefore the fragment stands alone in terms of functionality). The source code of the ground-truth implementation is given, firstly as simply the operator used on that line, and secondly in a C-like fashion (excluding braces). This C-like fashion is a programmatically generated translation of the source code of the custom language implementation, provided for ease of readability (due to the difficult-to-parse structure of the custom language). We then refer to the lines in this source code by line number. Fragments cannot contain end-of-block operators (used to indicate the end point of blocks started by the flow-control operators loop and conditional), nor can they contain the initial definition of the 2D canvas.

## A.6 FRAGMENTS EVALUATED FOR NN RECOGNISABILITY IN EXPERIMENT 2

The tables2425 describe the fragments (some of which contain requirements about variable dependencies) used in experiment 2.

## A.7 FULL RESULTS OF NN-SELECTED GUIDANCE FOR GP FROM EXPERIMENT 3

Tables 26,27,28 shows the success rate of the GP, if provided with hints by the neural network sets. Two sets of experiments are done on the array to array corpus, one on the canvas corpus. Most problems showed a success increase, including a number from both corpora which increased in success chance from 0% to a non-zero value. Two problems were made unfindable by the less-effective uniform fragment selection process, Iterative Difference and Trapezoid. As the baseline find rate was low, this does not represent a major drop in success, and may potentially simply be due to insufficient samples to determine the true success probability. We do not however reject the possibility that our approach has a negative effect on find-rates for certain problems. It was seen in Experiment 1 that some fragments, known to be present in the ground-truth implementation, decreased find-rates. It is possible that the neural networks correctly identified fragment presence, but that these degraded the GP's performance. It is, of course, also possible that the NN incorrectly estimated that a fragment was present when it was not, and that this erroneous hint harmed the GP.

| Problem | Baseline Find Rate |
|---|---|
| Abs | 8 % |
| ArrayLength | 100 % |
| ArrayToZero | 100 % |
| CumulativeAbsoluteSum | 0 % |
| CumulativeSum | 4 % |
| DivergentSequence | 58 % |
| FirstElementOnly | 28 % |
| Identity | 100 % |
| IndexParity | 100 % |
| IterativeDifference | 4 % |
| KeepEvens | 0 % |
| KeepNegatives | 0 % |
| KeepOdds | 0 % |
| KeepPositives | 60 % |
| Negative | 68 % |
| Pop | 30 % |
| RemoveFirstElement | 10 % |
| RetainFirstHalf | 0 % |
| Reverse | 58 % |
| ShiftLeft | 10 % |
| ShiftLeftZeroPadded | 40 % |
| ShiftRight | 0 % |
| ShiftRightLossy | 84 % |
| ShuffleZerosToBack | 100 % |
| Signum | 0 % |
| Sort | 0 % |
| SquareValues | 70 % |
| ToIterator | 100 % |
| | |
| Add | 24 % |
| Append | 0 % |
| ClipToMax | 18 % |
| ClipToMin | 4 % |
| ConstantAddition | 0 % |
| FillArray | 100 % |
| GreaterThan | 10 % |
| IterateFromStart | 98 % |
| LessThan | 8 % |
| MultiplesOf | 90 % |
| Multiply | 20 % |
| Subtract | 18 % |

Table 8: Find rates for a Genetic Algorithm as implemented above on the problems of the first corpus, a set of functions which take an input array of integers only (functions above dividing line) or an input array of integers and an integer. Both forms return a single array. n=30

| Problem | Baseline Find Rate |
|---|---|
| Square | 97 % |
| Hollow Square | 100 % |
| Parallelogram | 0 % |
| Hollow Parallelogram | 0 % |
| Mirrored Parallelogram | 7 % |
| Mirrored Hollow Parallelogram | 13 % |
| Right Triangle | 97 % |
| Hollow Right Triangle | 87 % |
| Mirrored Right Triangle | 60 % |
| Hollow Mirrored Right Triangle | 63 % |
| Inverted Right Triangle | 60 % |
| Hollow Inverted Right Triangle | 83 % |
| Inverted Mirrored Right Triangle | 100 % |
| Inverted Hollow Mirrored Right Triangle | 100 % |
| Isocelese Triangle | 0 % |
| Hollow Isocelese Triangle | 13 % |
| Inverted Isocelese Triangle | 47 % |
| Hollow Inverted Isocelese Triangle | 50 % |
| Rectangle With Empty Trapezoid | 3 % |
| Inverted Rectangle With Empty Trapezoid | 3 % |
| Obtuse Triangle | 3 % |
| Hollow Obtuse Triangle | 27 % |
| Mirrored Obtuse Triangle | 0 % |
| Mirrored Hollow Obtuse Triangle | 0 % |
| Inverted Obtuse Triangle | 0 % |
| Hollow Inverted Obtuse Triangle | 10 % |
| Inverted Mirrored Obtuse Triangle | 0 % |
| Hollow Mirrored Inverted Obtuse Triangle | 3 % |
| V Shape | 47 % |
| Trapezoid | 7 % |

Table 9: Find rates for a Genetic Algorithm (implementation described above) on the problems of the second corpus, a set of functions which take an integer size for the returned canvas, and must return the shape specified. n=30

| Line | Operator | As Code |
|------|----------|---------|
| 1 | Literal | variables[6] = 1; |
| 2 | Add | variables[7] = variables[0] + variables[6]; |
| 3 | Make Array | arrays[1] = new int[vars[7]] |
| 4 | Loop | for (variables[2]=0;variables[2]<variables[0];variables[2]++) |
| 5 | Read | variables[5] = arrays[0][variables[2]]; |
| 6 | Write | arrays[1][variables[2]] = variables[5]; |
| 7 | Endloop | |
| 8 | Write | arrays[1][variables[2]] = variables[1]; |

| Fragment | Success Rate |
|----------|--------------|
| 1 | 3% |
| 1, 2 | 27% |
| 1, 4 | 10% |
| 4 | 0% |
| 4, 5 | 0% |
| 4, 6 | 0% |
| 4, 8 | 0% |

Table 10: Fragments assessed from program "Append". Program's code listed, in C-like format, with operators listed ahead of each line for each of readability. Fragments then described, in reference to lines used followed by success rate using fragment as GP guidance (n=30)

| Line | Operator | As Code |
|------|----------|---------|
| 1 | Make Array | arrays[1] = new int[vars[0]] |
| 2 | Loop | for (variables[2]=0;variables[2]<variables[0];variables[2]++) |
| 3 | Literal | variables[5] = -1; |
| 4 | Read | variables[3] = arrays[0][variables[2]]; |
| 5 | Condition | if (variables[3]>0) |
| 6 | Else | else |
| 7 | Multiply | variables[3] = variables[3] * variables[5]; |
| 8 | Endloop | |
| 9 | Add | variables[4] = variables[4] + variables[3]; |
| 10 | Write | arrays[1][variables[2]] = variables[4]; |
| 11 | Endloop | |

| Fragment | Success Rate |
|----------|--------------|
| 1 | 0% |
| 1, 2 | 0% |
| 1, 3 | 0% |
| 1, 5 | 0% |
| 1, 6 | 0% |
| 2 | 0% |
| 2, 3 | 0% |
| 2, 4 | 3% |
| 2, 5 | 3% |
| 2, 6 | 0% |
| 3 | 0% |
| 3, 5 | 0% |
| 3, 6 | 0% |
| 3, 7 | 0% |
| 5 | 0% |
| 5, 6 | 3% |
| 6 | 0% |

Table 11: Fragments assessed from program "Cumulative Absolute Sum". Program's code listed, in C-like format, with operators listed ahead of each line for each of readability. Fragments then described, in reference to lines used followed by success rate using fragment as GP guidance (n=30)

| Line | Operator | As Code |
|------|----------|---------|
| 1 | Literal | variables[4] = 2; |
| 2 | Make Array | arrays[1] = new int[vars[0]] |
| 3 | Loop | for (variables[2]=0;variables[2]<variables[0];variables[2]++) |
| 4 | Read | variables[3] = arrays[0][variables[2]]; |
| 5 | Modulo | variables[5] = variables[3] % variables[4]; |
| 6 | Condition | if (variables[5]==variables[6]) |
| 7 | Write | arrays[1][variables[2]] = variables[3]; |
| 8 | Endloop | |
| 9 | Endloop | |

| Fragment | Success Rate |
|----------|--------------|
| 1 | 0% |
| 1, 2 | 6% |
| 1, 3 | 3% |
| 1, 5 | 3% |
| 2 | 3% |
| 2, 3 | 0% |
| 3 | 0% |
| 3, 4 | 0% |
| 3, 7 | 0% |

Table 12: Fragments assessed from program "Keep Evens". Program's code listed, in C-like format, with operators listed ahead of each line for each of readability. Fragments then described, in reference to lines used followed by success rate using fragment as GP guidance (n=30)

| Line | Operator | As Code |
|------|----------|---------|
| 1 | Literal | variables[6] = 2; |
| 2 | Divide | variables[3] = variables[0] / variables[6]; |
| 3 | Make Array | arrays[1] = new int[vars[3]] |
| 4 | Loop | for (variables[2]=0;variables[2]<variables[3];variables[2]++) |
| 5 | Read | variables[5] = arrays[0][variables[2]]; |
| 6 | Write | arrays[1][variables[2]] = variables[5]; |
| 7 | Endloop | |

| Fragment | Success Rate |
|----------|--------------|
| 1 | 0% |
| 1, 2 | 13% |

Table 13: Fragments assessed from program "Retain First Half". Program's code listed, in C-like format, with operators listed ahead of each line for each of readability. Fragments then described, in reference to lines used followed by success rate using fragment as GP guidance (n=30)

| Line | Operator | As Code |
|---|---|---|
| 1 | Literal | variables[7] = 2; |
| 2 | Make Array | arrays[1] = new int[vars[0]] |
| 3 | Loop | for (variables[2]=0;variables[2]¡variables[0];variables[2]++) |
| 4 | Subtract | variables[6] = variables[0] - variables[2]; |
| 5 | Subtract | variables[6] = variables[6] - variables[7]; |
| 6 | Read | variables[5] = arrays[0][variables[6]]; |
| 7 | Write | arrays[1][variables[2]] = variables[5]; |
| 8 | Endloop | |

| Fragment | Success Rate |
|---|---|
| 1 | 63% |
| 1, 2 | 80% |
| 1, 3 | 77% |
| 2 | 73% |
| 2, 3 | 60% |
| 3 | 63% |
| 3, 4 | 80% |
| 3, 7 | 77% |

Table 14: Fragments assessed from program "Reverse". Program's code listed, in C-like format, with operators listed ahead of each line for each of readability. Fragments then described, in reference to lines used followed by success rate using fragment as GP guidance (n=30)

| Line | Operator | As Code |
|---|---|---|
| 1 | Literal | variables[6] = 1; |
| 2 | Add | variables[8] = variables[0] + variables[6]; |
| 3 | Make Array | arrays[1] = new int[vars[8]] |
| 4 | Loop | for (variables[2]=0;variables[2]<variables[0];variables[2]++) |
| 5 | Add | variables[7] = variables[2] + variables[6]; |
| 6 | Read | variables[5] = arrays[0][variables[2]]; |
| 7 | Write | arrays[1][variables[7]] = variables[5]; |
| 8 | Endloop | |

| Fragment | Success Rate |
|---|---|
| 1 | 3% |
| 1, 2 | 13% |
| 1, 4 | 20% |
| 4 | 0% |
| 4, 6 | 0% |

Table 15: Fragments assessed from program "Shift Right". Program's code listed, in C-like format, with operators listed ahead of each line for each of readability. Fragments then described, in reference to lines used followed by success rate using fragment as GP guidance (n=30)

| Line | Operator | As Code |
|------|----------|---------|
| 1 | Literal | variables[6] = 2; |
| 2 | Add | variables[8] = variables[0] + variables[6]; |
| 3 | Make Array | arrays[1] = new int[vars[0]] |
| 4 | Subtract | variables[9] = variables[0] - variables[6]; |
| 5 | Loop | for (variables[2]=0;variables[2]¡variables[9];variables[2]++) |
| 6 | Add | variables[7] = variables[2] + variables[6]; |
| 7 | Read | variables[5] = arrays[0][variables[2]]; |
| 8 | Write | arrays[1][variables[7]] = variables[5]; |
| 9 | Endloop | |

| Fragment | Success Rate |
|----------|--------------|
| 1 | 80% |
| 1, 2 | 73% |
| 1, 3 | 63% |
| 1, 4 | 63% |
| 3 | 67% |

Table 16: Fragments assessed from program "Shift Right Lossy". Program's code listed, in C-like format, with operators listed ahead of each line for each of readability. Fragments then described, in reference to lines used followed by success rate using fragment as GP guidance (n=30)

| Line | Operator | As Code |
|---|---|---|
| 1 | Literal | variables[5] = 1; |
| 2 | Subtract | variables[1] = variables[0] - variables[5]; |
| 3 | Loop | for (variables[2]=0;variables[2]<variables[0];variables[2]++) |
| 4 | Loop | for (variables[3]=0;variables[3]<variables[1];variables[3]++) |
| 5 | Add | variables[6] = variables[3] + variables[5]; |
| 6 | Read | variables[4] = arrays[0][variables[3]]; |
| 7 | Read | variables[7] = arrays[0][variables[6]]; |
| 8 | Subtract | variables[8] = variables[4] - variables[7]; |
| 9 | Condition | if (variables[8]>0) |
| 10 | Write | arrays[0][variables[6]] = variables[4]; |
| 11 | Write | arrays[0][variables[3]] = variables[7]; |
| 12 | Endloop | |
| 13 | Endloop | |
| 14 | Endloop | |
| 15 | Make Array | arrays[1] = new int[vars[0]] |
| 16 | Loop | for (variables[2]=0;variables[2]<variables[0];variables[2]++) |
| 17 | Read | variables[5] = arrays[0][variables[2]]; |
| 18 | Write | arrays[1][variables[2]] = variables[5]; |
| 19 | Endloop | |

| Fragment | Success Rate |
|---|---|
| 1 | 0% |
| 1, 2 | 0% |
| 1, 3 | 0% |
| 1, 8 | 0% |
| 1, 15 | 0% |
| 1, 16 | 0% |
| 3 | 0% |
| 3, 8 | 0% |
| 3, 15 | 0% |
| 3, 16 | 0% |
| 3, 17 | 0% |
| 4 | 0% |
| 4, 6 | 0% |
| 8 | 0% |
| 8, 9 | 0% |
| 8, 15 | 0% |
| 8, 16 | 0% |
| 15 | 0% |
| 15, 16 | 0% |
| 16 | 0% |

Table 17: Fragments assessed from program "Sort". Program's code listed, in C-like format, with operators listed ahead of each line for each of readability. Fragments then described, in reference to lines used followed by success rate using fragment as GP guidance (n=30)

| Line | Operator | As Code |
|---|---|---|
| 1 | Make 2D Array | new 2DArray(size=variables[0]); |
| 2 | Literal | variables[6] = 2; |
| 3 | Divide | variables[4] = variables[0] / variables[6]; |
| 4 | Loop | for (variables[2]=0;variables[2]<variables[4];variables[2]++) |
| 5 | Loop | for (variables[3]=0;variables[3]<variables[4];variables[3]++) |
| 6 | Add | variables[7] = variables[2] + variables[3]; |
| 7 | Write to 2D | array[variables[7]][variables[3]]=1; |
| 8 | Endloop | |
| 9 | Endloop | |

| Fragment | Success Rate |
|---|---|
| 2 | 23% |
| 2, 3 | 30% |

Table 18: Fragments assessed from program "Mirrored Parallelogram". Program's code listed, in C-like format, with operators listed ahead of each line for each of readability. Fragments then described, in reference to lines used followed by success rate using fragment as GP guidance (n=30)

| Line | Operator | As Code |
|---|---|---|
| 1 | Make 2D Array | new 2DArray(size=variables[0]); |
| 2 | Literal | variables[6] = 2; |
| 3 | Divide | variables[4] = variables[0] / variables[6]; |
| 4 | Loop | for (variables[2]=0;variables[2]<variables[4];variables[2]++) |
| 5 | Add | variables[5] = variables[2] + variables[4]; |
| 6 | Write to 2D | array[variables[5]][variables[10]]=1; |
| 7 | Write to 2D | array[variables[2]][variables[4]]=1; |
| 8 | Subtract | variables[6] = variables[4] - variables[2]; |
| 9 | Write to 2D | array[variables[2]][variables[6]]=1; |
| 10 | Write to 2D | array[variables[5]][variables[6]]=1; |
| 11 | Endloop | |
| 12 | Literal | variables[8] = 1; |
| 13 | Subtract | variables[7] = variables[0] - variables[8]; |
| 14 | Write to 2D | array[variables[7]][variables[10]]=1; |

| Fragment | Success Rate |
|---|---|
| 2 | 13% |
| 2, 3 | 60% |
| 2, 12 | 10% |
| 12 | 13% |
| 12, 13 | 40% |

Table 19: Fragments assessed from program "Mirrored Hollow Parallelogram". Program's code listed, in C-like format, with operators listed ahead of each line for each of readability. Fragments then described, in reference to lines used followed by success rate using fragment as GP guidance (n=30)

| Line | Operator | As Code |
|------|----------|---------|
| 1 | Make 2D Array | new 2DArray(size=variables[0]); |
| 2 | Literal | variables[1] = 1; |
| 3 | Subtract | variables[4] = variables[0] - variables[1]; |
| 4 | Loop | for (variables[2]=0;variables[2]<variables[0];variables[2]++) |
| 5 | Write to 2D | array[variables[2]][variables[4]]=1; |
| 6 | Write to 2D | array[variables[5]][variables[2]]=1; |
| 7 | Write to 2D | array[variables[2]][variables[2]]=1; |
| 8 | Endloop | |

| Fragment | Success Rate |
|----------|--------------|
| 2 | 80% |
| 2, 3 | 90% |
| 2, 4 | 80% |
| 4 | 90% |
| 4, 6 | 63% |
| 4, 7 | 87% |

Table 20: Fragments assessed from program "Hollow Right Triangle". Program's code listed, in C-like format, with operators listed ahead of each line for each of readability. Fragments then described, in reference to lines used followed by success rate using fragment as GP guidance (n=30)

| Line | Operator | As Code |
|------|----------|---------|
| 1 | Make 2D Array | new 2DArray(size=variables[0]); |
| 2 | Literal | variables[3] = 1; |
| 3 | Subtract | variables[4] = variables[0] - variables[3]; |
| 4 | Loop | for (variables[2]=0;variables[2]<variables[0];variables[2]++) |
| 5 | Write to 2D | array[variables[2]][variables[4]]=1; |
| 6 | Write to 2D | array[variables[4]][variables[2]]=1; |
| 7 | Subtract | variables[5] = variables[0] - variables[2]; |
| 8 | Subtract | variables[5] = variables[5] - variables[3]; |
| 9 | Write to 2D | array[variables[2]][variables[5]]=1; |
| 10 | Endloop | |

| Fragment | Success Rate |
|----------|--------------|
| 2 | 67% |
| 2, 3 | 93% |
| 2, 4 | 80% |
| 4 | 67% |
| 4, 7 | 80% |

Table 21: Fragments assessed from program "Hollow Mirrored Right Triangle". Program's code listed, in C-like format, with operators listed ahead of each line for each of readability. Fragments then described, in reference to lines used followed by success rate using fragment as GP guidance (n=30)

| Line | Operator | As Code |
|------|----------|---------|
| 1 | Make 2D Array | new 2DArray(size=variables[0]); |
| 2 | Literal | variables[4] = 2; |
| 3 | Loop | for (variables[2]=0;variables[2]<variables[0];variables[2]++) |
| 4 | Multiply | variables[6] = variables[2] * variables[4]; |
| 5 | Subtract | variables[5] = variables[0] - variables[6]; |
| 6 | Loop | for (variables[3]=0;variables[3]<variables[5];variables[3]++) |
| 7 | Add | variables[7] = variables[3] + variables[2]; |
| 8 | Write to 2D | array[variables[7]][variables[2]]=1; |
| 9 | Endloop | |
| 10 | Endloop | |

| Fragment | Success Rate |
|----------|--------------|
| 2 | 23% |
| 2, 3 | 20% |
| 3 | 20% |

Table 22: Fragments assessed from program "Inverted Isoceles Triangle". Program's code listed, in C-like format, with operators listed ahead of each line for each of readability. Fragments then described, in reference to lines used followed by success rate using fragment as GP guidance (n=30)

| Line | Operator | As Code |
|------|----------|---------|
| 1 | Make 2D Array | new 2DArray(size=variables[0]); |
| 2 | Literal | variables[7] = 1; |
| 3 | Literal | variables[4] = 2; |
| 4 | Divide | variables[5] = variables[0] / variables[4]; |
| 5 | Loop | for (variables[2]=0;variables[2]<variables[0];variables[2]++) |
| 6 | Loop | for (variables[3]=0;variables[3]<variables[5];variables[3]++) |
| 7 | Subtract | variables[8] = variables[0] - variables[5]; |
| 8 | Divide | variables[8] = variables[8] / variables[4]; |
| 9 | Subtract | variables[8] = variables[8] - variables[3]; |
| 10 | Add | variables[9] = variables[8] + variables[7]; |
| 11 | Condition | if (variables[9]>0) |
| 12 | Subtract | variables[9] = variables[2] - variables[8]; |
| 13 | Condition | if (variables[9]>0) |
| 14 | Subtract | variables[9] = variables[0] - variables[8]; |
| 15 | Subtract | variables[9] = variables[9] - variables[2]; |
| 16 | Condition | if (variables[9]>0) |
| 17 | Write to 2D | array[variables[2]][variables[3]]=1; |
| 18 | Endloop | |
| 19 | Endloop | |
| 20 | Endloop | |
| 21 | Endloop | |
| 22 | Endloop | |

| Fragment | Success Rate |
|----------|--------------|
| 2 | 10% |
| 2, 3 | 10% |
| 2, 5 | 10% |
| 3 | 3% |
| 3, 4 | 0% |
| 3, 5 | 10% |
| 5 | 3% |

Table 23: Fragments assessed from program "Trapezoid". Program's code listed, in C-like format, with operators listed ahead of each line for each of readability. Fragments then described, in reference to lines used followed by success rate using fragment as GP guidance (n=30)

| Fragment | Description |
|---|---|
| Add | Simple addition. Requires the program to at some point contain an addition operation |
| +1 Offset Loop | Three Line Fragment
The first sets $Var1$ to 1
The second is a loop operator
The third requires an addition operator such that
$Var2 = loop\_iterator + Var1$ |
| Length -1 Loop | Three Line Fragment
The first sets $Var1$ to 1
The second requires an addition operator such that
$Var2 = input\_array\_size + Var1$
The third is a loop operator bounded to $Var2$ |
| Literal (2) | Requires a variable to be set to 2 |
| Loop | Requires the program to contain a loop |
| Loop Conditional | Two line fragment. The first line requires a loop
The second line requires a conditional $(var > 0)$ |
| Loop Read | Two line fragment. The first line requires a loop
The second line requires a read operation
such that the index read is the loop's iterator |
| Nonstandard Array | One line fragment. Requires the output array to be created,
with a size $Var1$ such that
$Var1$ is not the variable defaulting to input array size |
| Read | Requires the program to read from the input array |
| Add | Simple subtraction. Requires the program to at some point contain a subtract operation |

Table 24: Fragments used in experiment 2, corpus 1, to determine whether the NN can recognise their presence in a program's source code based on its behaviour. If multiple lines are required they are required to exist in order, but not necessarily consecutively. Variable numbering is not reflective of source-code implementation and for descriptive purposes only.

| Fragment | Description |
|---|---|
| Add | Simple addition. Requires the program to at some point contain an addition operation |
| Conditional | Greater than 0 operator. The program must contain an operator which executes a non-empty code block if a variable is greater than zero |
| Half | Two line fragment. The first sets a variable $Var1$ to the literal 2, The second assigns a variable to $desired\_output\_size/Var1$ |
| Half Loop | Three line fragment. The first sets a variable $Var1$ to the literal 2, The second assigns a variable $Var2$ to $desired\_output\_size/Var1$ The third defines a loop operator which runs from 0 to $Var2$ |
| Half Loop Depends | Four line fragment. The first sets a variable $Var1$ to the literal 2, The second assigns a variable $Var2$ to $desired\_output\_size/Var1$ The third defines a loop operator which runs from 0 to $Var2$ The fourth defines an operation setting a value on the 2D canvas, with the requirement that the X position of the point be logically dependent on the loop iterator |
| Loop Conditional | Two line fragment. The first requires a loop operator The second requires a conditional ($var > 0$) operator |
| Loop Draw | Two line fragment. The first requires a loop operator The second requires a 2D array write operation in which the X position drawn to depends logically on the loop's iterator |
| Loop Loop | Two line fragment. The program must have two loops (not necessarily nested) |
| Loop Loop Subtract | Three line fragment. The program must have two loops (not necessarily nested) It must then have a subtract operator |
| Draw Draw | Two line fragment. The program must have two draw-to-2D-array operators |

Table 25: Fragments used in experiment 2, corpus 2, to determine whether the NN can recognise their presence in a program's source code based on its behaviour. If multiple lines are required they are required to exist in order, but not necessarily consecutively. Variable numbering is not reflective of source-code implementation and for descriptive purposes only.

| Problem | Success Rate | Baseline |
|---------|--------------|----------|
| Abs | 73% | 7% |
| ArrayLength | 100% | 100% |
| ArrayToZero | 100% | 100% |
| CumulativeAbsoluteSum | 0% | 0% |
| CumulativeSum | 33% | 3% |
| DivergentSequence | 63% | 57% |
| FirstElementOnly | 100% | 27% |
| Identity | 100% | 100% |
| IndexParity | 100% | 100% |
| IterativeDifference | 3% | 3% |
| KeepEvens | 0% | 0% |
| KeepNegatives | 27% | 0% |
| KeepOdds | 10% | 0% |
| KeepPositives | 47% | 60% |
| Negative | 67% | 67% |
| Pop | 100% | 30% |
| RemoveFirstElement | 83% | 10% |
| RetainFirstHalf | 0% | 0% |
| Reverse | 77% | 57% |
| ShiftLeft | 80% | 10% |
| ShiftLeftZeroPadded | 83% | 40% |
| ShiftRight | 17% | 0% |
| ShiftRightLossy | 77% | 83% |
| ShuffleZerosToBack | 80% | 100% |
| Signum | 10% | 0% |
| Sort | 3% | 0% |
| SquareValues | 77% | 70% |
| ToIterator | 100% | 100% |
| Add | 37% | 23% |
| Append | 47% | 0% |
| ClipToMax | 43% | 17% |
| ClipToMin | 70% | 3% |
| ConstantAddition | 7% | 0% |
| FillArray | 100% | 100% |
| GreaterThan | 23% | 10% |
| IterateFromStart | 100% | 97% |
| LessThan | 17% | 7% |
| MultiplesOf | 87% | 90% |
| Multiply | 30% | 20% |
| Subtract | 43% | 17% |

Table 26: Success rates for problems of the 1st corpus, using the rarest-first guidance strategy, with the aggregate estimates from the feed-forward architecture network (n=30)

| Problem | Success Rate | Baseline |
|---|---|---|
| Abs | 27% | 7% |
| ArrayLength | 100% | 100% |
| ArrayToZero | 100% | 100% |
| CumulativeAbsoluteSum | 0% | 0% |
| CumulativeSum | 20% | 3% |
| DivergentSequence | 83% | 57% |
| FirstElementOnly | 83% | 27% |
| Identity | 100% | 100% |
| IndexParity | 100% | 100% |
| IterativeDifference | 0% | 3% |
| KeepEvens | 0% | 0% |
| KeepNegatives | 53% | 0% |
| KeepOdds | 10% | 0% |
| KeepPositives | 40% | 60% |
| Negative | 63% | 67% |
| Pop | 73% | 30% |
| RemoveFirstElement | 57% | 10% |
| RetainFirstHalf | 0% | 0% |
| Reverse | 53% | 57% |
| ShiftLeft | 43% | 10% |
| ShiftLeftZeroPadded | 63% | 40% |
| ShiftRight | 0% | 0% |
| ShiftRightLossy | 53% | 83% |
| ShuffleZerosToBack | 77% | 100% |
| Signum | 10% | 0% |
| Sort | 0% | 0% |
| SquareValues | 77% | 70% |
| ToIterator | 100% | 100% |
| Add | 23% | 23% |
| Append | 7% | 0% |
| ClipToMax | 33% | 17% |
| ClipToMin | 20% | 3% |
| ConstantAddition | 3% | 0% |
| FillArray | 100% | 100% |
| GreaterThan | 13% | 10% |
| IterateFromStart | 97% | 97% |
| LessThan | 30% | 7% |
| MultiplesOf | 93% | 90% |
| Multiply | 30% | 20% |
| Subtract | 17% | 17% |

Table 27: Success rates for problems of the 1st corpus, using the uniform guidance strategy, with the aggregate estimates from the feed-forward architecture network (n=30)

| | | |
|---|---|---|
| Square | 90% | 97% |
| HollowSquare | 100% | 100% |
| Parallelogram | 23% | 0% |
| HollowParallelogram | 7% | 0% |
| MirroredParallelogram | 43% | 7% |
| MirroredHollowParallelogram | 17% | 13% |
| RightTriangle | 93% | 97% |
| HollowRightTriangle | 97% | 87% |
| MirroredRightTriangle | 50% | 60% |
| HollowMirroredRightTriangle | 60% | 63% |
| InvertedRightTriangle | 67% | 60% |
| HollowInvertedRightTriangle | 57% | 83% |
| InvertedMirroredRightTriangle | 97% | 100% |
| InvertedHollowMirroredRightTriangle | 100% | 100% |
| IsoceleseTriangle | 10% | 0% |
| HollowIsoceleseTriangle | 43% | 13% |
| InvertedIsoceleseTriangle | 37% | 47% |
| HollowInvertedIsoceleseTriangle | 33% | 50% |
| RectangleWithEmptyTrapezoid | 3% | 3% |
| InvertedRectangle | 10% | 3% |
| obtuseTriangle | 23% | 3% |
| hollowObtuseTriangle | 53% | 27% |
| mirroredObtuseTriangle | 0% | 0% |
| mirroredHollowObtuseTriangle | 0% | 0% |
| invertedObtuseTriangle | 0% | 0% |
| hollowInvertedObtuseTriangle | 17% | 10% |
| invertedMirroredObtuseTriangle | 7% | 0% |
| hollowMirroredInvertedObtuseTriangle | 7% | 3% |
| VShape | 40% | 47% |
| Trapezoid | 0% | 7% |

Table 28: Success rates for problems of the 2nd corpus, using the uniform guidance strategy, with the aggregate estimates from the convolutional architecture network (n=30)

