# OpenReview forum: "Neurally Guided Genetic Programming for Turing Complete Programming by Example"
_ICLR.cc/2021/Conference — Reject_

### Official Review · AnonReviewer2 · 2020-10-26
**A possibly useful idea, requires better explanations and experiments**

**Rating:** 4
**Confidence:** 4

**Review:**

This paper proposes a genetic programming approach for programming-by-example systems, where the genetic search is seeded with fragments that are predicted by a neural network to be relevant to the input-output examples. This is shown to solve more synthesis problems compared to un-aided genetic programming for array-to-array programs and canvas drawing programs.

The neural network is a classifier that given a set of input-output examples and a program fragment predicts whether the fragment could occur in the solution to the synthesis problem. The fragments above a threshold are used for seeding the genetic search and candidates that do not include them are penalized during the search. An initial genetic search is performed on simpler problems to prepare the training set for the classifier. I found the description of the process for generating the training set quite vague. From what I understand, the useful fragments from candidates generated during the initial genetic search are identified manually and then it is augmented with a further set of manually chosen fragment. Some noisy fragments are then added randomly to prepare negative examples. The extent of the manual efforts here is substantial and requires domain knowledge from the developer. In contrast, many neural models for synthesis work entirely off of randomly sampled programs.

The experimentation is not explained clearly. The term "find rates" is used without defining. It appears from Table 1 that it is some way of measuring the success of genetic search (but what and how it computed is not clear). In Section 4.2, it is said that "all programs with a 90% find rate or better ..." are used. What is the find rate of a program and how does it relate to the find rate of the search procedures?

Experiment 3 shows the benefits of using the classifier to seed the search. What is the relation between the problems whose fragments are used for training the classifier and the problems used in this experiment? Is the former a subset of the latter?

From the applicability point of view, what is the time take by a successful genetic search across the problems? The rarest preferred fragment assumes existence of other problems to rank order the fragments. This is an impractical requirement to the user, who would be interested in only her problem and cannot be expected to supply related problems for ranking purposes.

From the point of view of baselines, there are two types of baselines missing. One is a neural method for straightforward program generation from examples; any suitable one from the cited papers can be used. Another is a non-neural selection method for fragments; a simple mining method can be applied to the data from the correct programs from the initial genetic search. These will help properly understand the benefits of the respective contributions.

Please label the sub-tables in Table 1 and 2. Can you spell out how the distance D for identifying repulsors is computed, for instance, if some edit distance is used then which one?

---

> ### Author Response · Authors · 2020-11-17
> **Response to review**
>
> This review seeks a set of clarifications, and also comments on potential baseline comparisons. We are very happy to clarify the exposition in the paper in each of the requested areas. We also respond to the main questions as follows:
>
> **"What is the relation between the problems whose fragments are used for training the classifier and the problems used in this experiment? Is the former a subset of the latter?"**
>
> They are indeed a subset. We can alter the presentation of the tables to better reflect this, and to organise into a seeds (roughly equivalent to training) and testing problem set, as well as correcting imprecise / unclear language identified in the review.
>
> **"This is an impractical requirement to the user, who would be interested in only her problem"**
>
> We tackle the context of a system in which multiple problems will need to be faced by a shared system, such as an online server communicating with many users, across which it would be natural to share successes in the back-end implementation of the system. The user would therefore only see her solution processed, but any success would be fed back with fragments extracted to aid requests of other users. Our approach also supports scenarios where no initial fragments exist by using simple genetic programming, with no guidance, to solve simpler problems which are informative for more complex ones.
>
> **"From the point of view of baselines, there are two types of baselines missing."**
>
> As noted in response to AnonReviewer4, we are not aware of any other neural synthesis approaches in the literature which are able to tackle the extreme search sizes involved in a full Turing-complete language (whose structure has that of a general-purpose language, as opposed to a task-specific DSL). Our work exceeds the spaces tackled by, say, DeepCoder by tens of orders of magnitude (10^10 vs 10^110). As noted in our response to AnonReviewer4, we would either need to (1) construct an infeasibly large neural network to have any chance of capturing relevant features of this search space, and collect an impossibly large number of samples of program space to gain any useful training, or (2) reduce the complexity of our DSL to be tractable to the DeepCoder approach. We argue that the former option is simply not possible with current technology, and the latter option would not yield an informative like-for-like result.
>
> With regard to other formatting and discussion issues, we very much appreciate the suggestions and will update our paper appropriately; we will also provide further details on the code distance metric.

---

> > ### Comment · AnonReviewer2 · 2020-11-21
> > **Additional clarifications**
> >
> > Thank you for your response.
> >
> > Can you respond to the other points raised in my review: the time required by successful genetic search and comparison to a non-neural baseline for selection of fragments?

---

> > > ### Author Response · Authors · 2020-11-24
> > > **Additional response**
> > >
> > > Time taken for a genetic search varies between a few minutes, in the case of success, to a little over 4 hours in the worst case. We were not optimising particularly for clock-time in this research, so a range of optimizations are likely to be possible here -- in particular we note that genetic search procedures are highly parallelizable and so would benefit from a larger number of processors / machines.
> > >
> > > We do not have an assessment for non-neural selection of fragments, but believe that the difference between outcomes based on rarity-first selection, derived from neural predictions, demonstrates that the output of the neural networks can be used to exceed a pure random selection process.

---

### Official Review · AnonReviewer1 · 2020-10-28
**Interesting work, but should have more compelling experiments given the background of related work**

**Rating:** 5
**Confidence:** 4

**Review:**

Summary:

This work considers training neural networks to provide "hints" (suggested lines of code) as a way of augmenting genetic programming systems. At test time, a neural network conditions on input-output examples and learns to predict 1-2 lines of code which are present in a program satisfying those input-outputs. At train time, the system learns in a self supervised manner by bootstrapping off previously discovered programs, augmenting them to produce a large corpus of similar problems, and then training the network to predict lines of code present in this augmented data set, similar to DeepCoder.

Overall the method is sound, but I have two objections which caused me to lean toward reject:

+ First, the experiments seem small-scale, and even on these small domains the absolute performance gain is modest (for one-dimensional array manipulation, their method solves 55% vs 38% with vanilla genetic programming, as measured on 40 synthesis problems). This is at odds with the claim that the "approache is scalable to the search space of Turing-complete languages" - why not try on a standard benchmark, such as SyGuS? The promise of the proposed method hinges on its raw performance as well as its general applicability, and at present the experiments do not highlight these two factors.
+ Second, there is prior work on using deep learning to accelerate stochastic search over program spaces, indeed even over spaces as rich as arbitrary assembly code, e.g. "Learning to superoptimize programs" (ICLR 2017). How does your method compare? This prior work showed compelling results on a challenging standard benchmark - can your method do the same?

To be clear, not every paper has to beat standard benchmarks --- new good ideas that are carefully investigated are also good, and Experiments 1-2 do a great job of showing experimentally the different elements of the approach in action. But there is enough existing work (see "missing citations") in the conceptual neighborhood of this submission that I really think they should compare on some standard-ish benchmarks, because the approach on its own seems too similar to prior works.

minor concerns:
- page 2: what do you mean by neural synthesis either drawing from the target language by "sampling at a uniform interval or at random"?
- how sensitive is the performance of the algorithm to the small details of the genetic programming set up?

missing citations:
- "Learning to superoptimize programs" (ICLR 2017 - see above)
- "Augmenting Genetic Algorithms with Deep Neural Networks for Exploring the Chemical Space" (ICLR 2020)
- "Learning Fitness Functions for Genetic Algorithms" (2019 - could this be combined synergistically with your work? where you learn the best fragments to include and they learn the fitness function?)
- "Write, Execute, Assess: program synthesis with a REPL" (NeurIPS 2019 - can this also be combined, whereby a mutation policy and fitness value function are learned through RL?)

---

> ### Author Response · Authors · 2020-11-17
> **Response to review**
>
> The main concerns in this review are on the scale of our experiments, and existing benchmarks:
>
> **"the experiments seem small-scale...the promise of the proposed method hinges on its raw performance as well as its general applicability"**
>
> While the number of lines of code involved in our experiments reaches 20, the experiments involve search spaces of up to 10^100 permutations of operators due to the complexity of the target language. This is many orders of magnitude larger than any other work we are aware of in the neural source synthesis community; we do therefore think that this comment reflects the complexity of our work. On the point of general applicability, we show results on two very different domains: classic array-to-array programs up to and including sorting algorithms, and canvas drawing programs up to and including drawing a trapezoid. We use the same target language across both problem domains, and use only 10 I/O examples to successfully synthesise programs -- including the synthesis of loops and conditional branch statements. We are unaware of any other work which approaches this level of generality.
>
> **"why not try on a standard benchmark, such as SyGuS?"**
>
> We do not believe that this benchmark is representative of our domain. We operate in the domain of programming by example, rather than syntax-guided synthesis, so the SyGuS benchmark does not appear to be an immediately applicable comparison point. We are unaware of any suitable benchmark suites which would represent a similar level of search space for comparison; we have therefore used common array transformation programs, plus a set of programs from existing work by So & Oh.
>
> **"there is prior work on using deep learning to accelerate stochastic search over program spaces"**
>
> The problems studied in "Learning to superoptimize programs" are solvable without use of a Turing-Complete language ("Synthesis of Loop-Free Programs"), and therefore not representative of the kind of challenge we address.
>
> With regard to **missing citations**, we thank the reviewer for these suggestions and would be happy to include those which are published. Specifically:
>
> -"Learning to superoptimize programs" does share similarity with our work in that it uses an NN to guide search through program space; however, it differs considerably as it only aims to refactor an existing program, not create a new one from scratch.
>
> -"Augmenting Genetic Algorithms with Deep Neural Networks for Exploring the Chemical Space" employs its NN in the same capacity as we use novelty search (a technique we found boosted our baseline success very strongly); it is therefore partially related but methodoligically distinct
>
> -"Learning Fitness Functions for Genetic Algorithms" We appreciate this suggestion, we are not aware of a peer-reviewed version of this paper and so have not cited it; we would be happy to cite and discuss a peer-reviewed iteration if available.
>
> -"Write, Execute, Assess: program synthesis with a REPL" focuses on a different problem domain, but we agree it is interesting work which we could build on in the future.

---

### Official Review · AnonReviewer4 · 2020-10-28
**Good idea, impressive synthesis results, but lacks non-GP baselines and some evaluation details unclear**

**Rating:** 5
**Confidence:** 5

**Review:**

The main idea of the paper is using neural networks to provide "starting hints" to a program synthesizer that is based on genetic programming. Specifically, the network predicts one or two lines that must be present in the desired program given the I/O examples for the task. These lines are used to initialize the GP process instead of an empty program.

The core idea resembles DeepCoder, with two (important) differences: (a) the network predicts a code fragment of 1-2 lines instead of a distribution over operators, (b) the subsequent synthesis algorithm is a GP process rather than symbolic search. However, there's no comparison with the DeepCoder approach, either on this paper's DSL/dataset, or on DeepCoder's. I appreciate the authors' argument that the two DSLs significantly differ in expressiveness. However, most programs in this dataset have <10 lines, and comparison with enumerative or neural-guided synthesis approaches would be both feasible and informative for them.

In fact, the only baseline comparison is with vanilla GP. As such, the paper is definitely valuable for the GP program synthesis community, but less valuable to the ICLR neural program synthesis community at large.
I also some questions regarding the validity of experimental setup and suggestions on the presentation clarity (see below). All of these prevent me from raising the score above acceptance at this time even though I find the main idea promising and the end-to-end evaluation results impressive.

### Evaluation questions

The experimental setup is a bit unorthodox. The authors present individual components of their approach simultaneously as both (a) ablation experiments and (b) stages of their end-to-end pipeline. According to this presentation, the end-to-end pipeline operates as follows:
1. Start with a subset of easy tasks. Identify 10 best code fragments that, if provided, increase the find rate of GP (via brute force enumeration of all fragments).
2. Generate a synthetic train/test set for the classification problem "IO -> Fragment Presence" for each fragment.
3. Train a classification NN for each fragment.
4. For all tasks (including the ones from step 1), use the trained NNs to select the code fragments to provide as hints to GP. Measure the find rate of GP.

The pipeline view would be better appreciated in the beginning of Section 3, as a high-level overview of the whole system. Without it, different experiments now clearly map to different research questions:
- Experiment 3 is the "main" end-to-end performance/generalization evaluation.
- Experiment 1 is the "oracle" version of it where the fragment prediction component is replaced by brute-force enumeration of all valid fragments from ground truth ("How good we could possibly get?").
- Experiment 2 is the evaluation of classification accuracy of that individual component.

After this perspective on evaluation is better established, two questions arise.

First, the inclusion of Experiment 1 tasks into the final measurement is odd - they are indirectly part of the "training" data in this setup because they influence the selection of useful fragments. It's unclear which of the newly-solved problems are "training" and which are "testing" without inspecting Tables 26-27 line-by-line. If Experiment 3 did not involve any tasks from Experiment 1, the new average find rate would better inform the reader how the hint fragments - and the approach as a whole - generalizes across tasks. Which is the main goal of this end-to-end experiment.

Second, when comparing Tables 26-27 in the appendix with the corresponding "oracle" numbers in Table 1, I get confused. For example, the Append task reaches a maximum of 27% find rate for the best-performing fragment in Table 1. However, the paper's approach predicts a fragment that facilitates a find rate of 47%. If Experiment 1 evaluated every possible valid fragment from the ground truth program, how could the approach find a fragment that's better than the best one? I'm probably misunderstanding something in the experimental setup.

### Presentation questions

In addition to Section 4 advice above, I suggest restructuring the technical sections of the paper (i) from high-level to low-level, (ii) with motivating examples, (iii) with a system overview figure in the beginning. As written, Section 3 presents a lot of technical detail and high-level motivations are hard to notice and appreciate until one reads the entire paper.
In contrast, So & Oh, whose tasks the authors re-use in this work, start with task/program examples and give a much more formal presentation of the entire synthesis system.

Sections 3.2-3.3 are quite dense. They describe almost the entire system in prose, with quite a bit of technical detail. Instead, this section would benefit from presenting it as two algorithms in pseudocode - one for the GP synthesis and one for the end-to-end system for training NN for fragment classification and using its predictions to drive GP.

---

> ### Author Response · Authors · 2020-11-17
> **Response to review**
>
> The central concerns within this review are on baseline comparisons, and an apparent discrepency in two tables.
>
> **"I appreciate the authors' argument that the two DSLs significantly differ in expressiveness. However, most programs in this dataset have <10 lines, and comparison with enumerative or neural-guided synthesis approaches would be both feasible and informative for them."**
>
> We disagree with this criticism; we contend that a comparison with DeepCoder would unfortunately not be informative. The DeepCoder approach works by uniformly sampling program space to train the model. Due to the differences in search space size (DeepCoder operates on spaces of size 10^10, while our search size is 10^100, with potentially 10^40 unique functionalities according to our assessment) we would either need to (1) construct an infeasibly large neural network to have any chance of capturing relevant features of this search space, and collect an impossibly large number of samples of program space to gain any useful training, or (2) reduce the complexity of our DSL to be tractable to the DeepCoder approach. We argue that the former option is simply not possible with current technology, and the latter option would not yield an informative like-for-like result. While the GP baseline that we use is indeed relatively simple, we are therefore unable to see another viable option here.
>
> **"If Experiment 1 evaluated every possible valid fragment...how could the approach find a fragment that's better than the best one"**
>
> We can clarify this apparent discrepency: in experiment 1, the fragments are drawn from a singular implementation of the ground truth program that we are searching for. In experiment 3, our approach has access to all fragments from across all of the successfully-found programs. We therefore hypothesise that fragments from different programs drove the additional success of the Append program, and can confirm from logs that there existed multiple successful runs for that problem which used fragments derived from other seed programs. We will clarify this in the paper.
>
> Aside from these points, we are grateful for the suggestions on improving the technical descriptions of the paper including further explanatory paragraphs, a summary diagram of the approach, and a more algorithmic treatment of the core mechanics of the approach -- we will implement these changes.

---

### Decision · Program_Chairs · 2021-01-07
**Final Decision**

**Decision:**

Reject

**Comment:**

There are some interesting ideas in this paper, but I agree with reviewers that without a comparison to existing work, it is hard to place this work in its proper context. The authors make several arguments in dismissing the need for side-by-side comparisons, but I do not find these arguments convincing.
* First, the authors argue that there are no suitable benchmarks for them to compare, and that in particular SyGuS benchmarks would not be suitable because they are dealing with a different problem. I disagree. There are 2 tracks in SyGuS specifically for programming-by-example problems, one for string manipulations and one for bit-vector programs. I think the string manipulation problems would be a good match for this technique.
* The authors also argue that their technique is so much more general than prior techniques that a side-by-side comparison would be unfair. However, their most complex benchmark, sorting, has been somewhat of a standard benchmark in the program synthesis community for about a decade now. And while a lot of recent synthesis work has focused on domain specific languages, many systems starting with Sketch and continuing with Myrth and Synquid were turning complete. Turing completeness can make a big difference if you are trying to synthesize verified code, but in the context of programming-by-example, turning completeness does not really present any fundamental challenges.

I am willing to believe that this technique is more scalable than existing techniques, so that while existing techniques may do better than this technique when synthesizing for small languages, this technique would surpass them when applied to a bigger language. But if that's the argument that the authors want to make I would like to see some evidence, and ideally some quantitative data as to how big a language would have to be before this technique wins out.